# PHYSICS OF LANGUAGE MODELS: PART 2.1, GRADE-SCHOOL MATH AND THE HIDDEN REASONING PROCESS

[EXTENDED ABSTRACT]*

**Tian Ye**
FAIR at Meta and Carnegie Mellon University
`sigma648@meta.com` and `tye2@andrew.cmu.edu`

**Zicheng Xu**
FAIR at Meta (now Google Research)
`zichengxu@meta.com` (now `zichengxu@google.com`)

**Yuanzhi Li**
Mohamed bin Zayed University of AI
`Yuanzhi.Li@mbzuai.ac.ae`

**Zeyuan Allen-Zhu**
FAIR at Meta
`zeyuanallenzhu@meta.com`

## ABSTRACT

Recent advances in language models have demonstrated their capability to solve mathematical reasoning problems, achieving near-perfect accuracy on grade-school level math benchmarks like GSM8K. In this paper, we formally study how language models solve these problems. We design a series of controlled experiments to address several fundamental questions: (1) Can language models truly develop reasoning skills, or do they simply memorize templates? (2) What is the model's hidden (mental) reasoning process? (3) Do models solve math questions using skills similar to or different from humans? (4) Do models trained on GSM8K-like datasets develop reasoning skills beyond those necessary for solving GSM8K problems? (5) What mental process causes models to make reasoning mistakes? (6) How large or deep must a model be to effectively solve GSM8K-level math questions?

Our study uncovers many hidden mechanisms by which language models solve mathematical questions, providing insights that extend beyond current understandings of LLMs.

## 1 INTRODUCTION

The field of language models has made significant progress in recent years. Large models like GPT-4 (OpenAI, 2023) have shown initial signs of general intelligence (Bubeck et al., 2023) — even argued to have reached L2 or L3-level of intelligence (Allen-Zhu & Xu, 2025), while smaller models have demonstrated good reasoning abilities by solving challenging coding/math problems (Li et al., 2023; Gunasekar et al., 2023; Liu et al., 2023).

---

*The first six papers in the *Physics of Language Models* series were presented as a two-hour tutorial at ICML 2024 in Austria (`youtu.be/yBL7J0kgldU`). A one-hour deep dive into Part 2.1 is available at `youtu.be/bpp6Dz8N2zY`. Full and future editions of Part 2.1, including code release, are available at `physics.allen-zhu.com`, `github.com`, and `ssrn.com/abstract=5250629`.

In this paper, we focus on the ability of small language models to solve grade-school math problems. Unlike previous works that empirically push the accuracy of models on grade-school math benchmarks like GSM8K (Cobbe et al., 2021) and its augmentations (e.g., Liu et al. (2023); Zhang et al. (2024)), we take a more principled approach. We aim to understand the following fundamental questions:

1. How do language models learn to solve grade-school level math problems? Do they just memorize templates, or do they learn reasoning skills similar to humans? Or do they discover new skills to solve the problems?

2. Do models trained *solely* on grade-school math problems only learn to solve these problems, or do they develop some more general intelligence?

3. How small can a language model be while still solving grade-school math problems? Is depth (number of layers) more important than width (number of neurons per layer), or does only size matter as suggested by practitioners (Kaplan et al., 2020)?

These questions are fundamental to understanding the intelligence of language models. To study them, it might seem tempting to start with a pre-trained model and fine-tune it on existing datasets like GSM8K or GPT-4 augmented ones (e.g., Liu et al. (2023); Zhang et al. (2024)). However, this approach has significant limitations:

- DATA CONTAMINATION. The pretrain data of existing models mostly come from publicly available internet (Gao et al., 2020), which is a pile of mess. We do not know how many math problems are included or their structures. There is *significant concern regarding* whether the GSM8K benchmark has been *leaked to language models' training datasets* (Zhang et al., 2024). Even if the exact data is not, the pre-trained model might have seen almost identical questions (e.g., the same problem with different numbers). Thus, this approach cannot answer questions 1-3. We do not know whether a model truly learns the reasoning skills or it simply memorizes problem templates during training. Therefore, we **need full control over the model's pretrain data** and must train a language model from scratch. This point has been reiterated recently in (Allen-Zhu & Li, 2024; 2025a).

- SOLUTION DIVERSITY. The existing fine-tuning data, such as the GSM8K training set, contains only 7.5K grade-school math problems, which is insufficient to train a model from scratch. Although recent works use GPT-4 to augment GSM8K, this is not enough for our purpose. GPT-4 augmented problems might be biased towards a small number of solution templates, since the original GSM8K data has very few (obviously, at most 8K) solution templates. **We need a much larger, more diverse set of grade-school math problems**.

With these points in mind, we introduce our framework to generate a large set of diverse grade-school math (GSM) problems and use the dataset to train (from scratch) and test a GPT2-like language model. In the framework, we focus on the "logical reasoning" aspect of grade-school math problems, which involves the dependency of parameters in the problem statement, such as "Alice's apple is three times the sum of Bob's orange and Charles's banana." We use synthetic sentences to reduce the difficulty arising from *Common Sense*, like "a candle burned for 12 hours at 1 inch per hour" (implying the candle is reducing in length). We also *intentionally remove* the difficulty from pure arithmetic: we only consider integers $\mathrm{mod}\,23$.[1]

Moreover, our framework ensures that the generated math problems are highly diverse and do not come from a small subset of templates. Even ignoring all the arithmetic, English, variable names, and unused parameters, our problems still have more than 90 trillion solution templates (see Proposition 2.2), much larger than the size of GPT2-small (100M). Thus, language models **cannot** solve the math problems in our case **by simply memorizing** the solution templates.

---

[1]The conclusions of this paper remain valid if one replaces 23 with, for instance, 2003. However, for a better-controlled experiment, we wish to *separate reasoning from arithmetic*. For instance, if a model fails, we want to ensure it is *not* due to an arithmetic error — after all, memorizing the multiplication table for 23 integers is trivial for even a small model. On a separate note, there is a rich literature studying how well language models can learn arithmetic and length generalization; see Zhou et al. (2023); Jelassi et al. (2023); Lee et al. (2023) and the references therein. Modern language models are also equipped with calculator plugins. Thus, it might not be the language model's job to perform complex arithmetic (such as 20-digit multiplications) anyway.

In this paper, we use the GPT2 model (Radford et al., 2019), but replace its positional embedding with rotary embedding (RoPE) (Su et al., 2021; Black et al., 2022). We still call it GPT2 for brevity. We summarize our main contributions:

– RESULT 2. We demonstrate that the GPT2 model, pretrained on our synthetic dataset, not only achieves 99% accuracy in solving math problems from the same distribution but also generalizes to out-of-distribution problems, such as those requiring *longer reasoning lengths* than any seen during training. This is similar to length generalization in arithmetic (Anil et al., 2022; Jelassi et al., 2023), however, in our case, the model **has never seen *any*** training example of such reasoning length. This signifies that the model can genuinely learn reasoning skills instead of memorizing solution templates.

– RESULT 3. Crucially, the model can learn to generate shortest solutions, almost always avoiding *unnecessary* computations. This suggests that the model *formulates a plan* before it generates, avoiding computing any quantities not needed towards solving the underlying math problem.

– RESULT 4. We examine the model's internal states through probing, introducing six probing tasks to elucidate *how* the model solves math problems. For instance, we discover the model (mentally!) preprocesses the full set of necessary parameters before it starts any generation. Likewise, humans also do this preprocess although we write this down on scratch pads.

– RESULT 5. Surprisingly, the model also learns *unnecessary, yet important* skills after pretraining, such as all-pair dependency. Before any question is asked, it already (mentally!) computes with good accuracy which parameters depend on which, even though *some are not needed for solving the math problem*. Note that computing all-pair dependency is a skill **not needed** to fit all the solutions in the training data. To the best of our knowledge, this is the first evidence that a language model can **learn useful skills beyond** those necessary to fit its pretraining data.[2] This may be a preliminary signal of **where the G in AGI** can come from.[3]

– RESULT 6. We explain *why* mistakes occur. For instance, the model makes systematic errors that can be explained by probing its internal states. Sometimes, these mistakes can be predicted before the model generates answers, making them independent of the random generation process. We connect this to practice, noting that GPT-4/4o also makes similar errors (though we cannot probe their internal states).

– RESULT 7+8. The depth of the language model is crucial for its reasoning ability. For example, a 16-layer, 576-dim transformer solves harder problems (in reasoning length) than a 4-layer, 1920-dim one, despite the latter being twice as large. This holds even when Chain-of-Thought (CoT) is used. We explain this necessity in depth by the complexity of the mental processes involved. We advocate for the use of controlled, synthetic data as a more principled approach to derive such claims, contrasting with predictions like "only size matters" based on training loss using internet pretrain data (Kaplan et al., 2020).

While we refrain from overstating that our findings directly apply to foundation models like GPT-4 or more challenging mathematical reasoning tasks, we believe our work significantly advances the understanding of how language models develop their mathematical reasoning skills, and this **has to be done in a way different from pushing benchmarks**.

## 2 RESULT 1: DATA GENERATION

**Motivation.** Recall a standard grade-school math problem in the GSM8K dataset (Cobbe et al., 2021) looks like:

> Betty is saving money for a new wallet which costs 100. Betty has only half of the money she needs. Her parents decided to give her 15 for that purpose, and her grandparents twice as much as her parents. How much more money does Betty need to buy the wallet?

---

[2]In our case, one can solve all the math problems without computing all-pair dependency. Our pretraining data never includes such information — all the solutions only compute necessary variables.

[3]Indeed, the skill to sort relationships among in-context objects is a general skill, which may lead to — via instruction fine-tuning — skills for solving other tasks, such as discovering causal relationships, determining the influence of parameter changes, etc.

Figure 1: Structure and dependency graph corresponding to the op $= 7$ easy example in (2.1) and (2.2). Dependencies from abstract parameters are drawn in **red**, and from instance parameters are in **black**.

This problem involves multiple parameters whose values are connected through various equalities, such as "Betty's current money = 0.5 × cost of the wallet" and "money given by grandparents = 2 × money given by parents." Motivated by this, we build a GSM8K-like math dataset through a synthetic generation pipeline that captures the dependencies of parameters. We wish to capture at least the following three types of dependencies.

1. Direct dependency ($\heartsuit$): such as $A = 5 \times (X + Y)$, so $A$ can be computed after $X$ and $Y$.

2. Instance dependency ($\spadesuit$): such as "every classroom has X chairs, and there are Y classrooms." Here, the model must infer the total number of chairs by multiplying X by Y.

3. Implicit dependency ($\clubsuit$): such as "Bob has 3 times more fruits than Alice. Alice has 3 apples, 4 eggs and 2 bananas." Here, the model must learn that apples and bananas are fruits and egg is not, and "Alice's fruits" is an abstract parameter derived from the problem statement.

## 2.1 STEP 1: GRAPH CONSTRUCTION AND PROBLEM GENERATION

**Hierarchical categorization.** We use a layered structure of *categories*, each contains possible *items*. For instance, categories = (School, Classroom, Backpack) has three layers; category School = {Central High, Riverview High, ...}; category Classroom = {Dance Studio, Film Studio, ...}; category Backpack = {School Daypack, Messenger Backpack, ...}. We prepare 4 predefined hierarchical categorizations, each with 4 layers and 100 items/layer; this represents the world knowledge.

**Structure graph.** In each math problem, only specific items exist, leading to a *structure graph* that outlines what sub-items can appear under what item, see Figure 1 (left). For instance,

- Connecting Dance Studio and School Daypack with an edge signifies an *instance parameter*, "the number of school daypacks in each dance studio," which is a quantifiable variable that can be assigned.[4] This captures the instance dependency ($\spadesuit$) as mentioned above.

- *Abstract parameters*, like "the total number of classrooms in Central High," cannot be assigned and are excluded from the structure graph. They reflect implicity dependency ($\clubsuit$) .

*Remark* 2.1. Rather than using simple objects like *Alice's apple* or fake items like *Items A/B/C/D*, this structure allows us to describe abstract parameters and adds 2 levels of complexity to the data:

- The model must implicitly learn English concepts, such as a classroom category includes 100 different classroom types. These concepts cannot be derived from individual math problems, as only a limited selection of classrooms will be mentioned in each problem.

- The model is required to hierarchically access multiple items to calculate abstract parameters, as opposed to a straightforward retrieval of "Alice's apple" in the context.[5]

---

[4]Even though Central High and Rivierside High can both have (possibly multiple) Dance Studios, for simplicity, we assume that each Dance Studio has the same number of School Daypacks.

[5]For example, the total number of backpacks in Riverview High in Figure 1 is calculated as $ip_1 \times ap_1 + ip_2 \times ap_2$ where $ip_1$ = "Riverview High's number of Dance Studios", $ip_2$ = "Riverview High's number of Film Studios", $ap_1$ = "each Dance Studio's number of Backpacks", and $ap_2$ = "each Film Studio's number of Backpacks", with $ip_1, ip_2$ being instance parameters and $ap_1, ap_2$ abstract parameters. Here, the model must not only retrieve $ip_1, ip_2$ but also compute $ap_1, ap_2$ hierarchically.

**Dependency graph.** The *dependency graph* is a directed acyclic graph that outlines the dependency among parameters. For each *instance parameter*, we choose a random set of (up to 4) parameters it can depend on — including possibly a special vertex RNG representing a random number generator. For instance, if "[param A] is $X$ more than the difference of [param B] and [param C]" for $X$ being randomly generated, then we draw edges from B, C and RNG to parameter A. The dependency of abstract parameters is implied by the dependency of instance parameters. This captures direct dependency ($\heartsuit$) as mentioned above. We give an examples on the right side of Figure 1, and details for how we randomly generate such dependency graph are in the full paper.

**Problem generation.** The *problem* is articulated by describing the dependency graphs in English, one sentence for each instance parameter.[6] (Abstract parameters are not described because they are inherited by the structure graph.) We **randomly permute** the sentence ordering to further increase difficulty. A parameter is selected and asked with a question in the end (or at the beginning). Below is an easy example corresponding to Figure 1; a harder example is in Figure 5.

> **(Problem - Easy)** The number of each Riverview High's Film Studio equals 5 times as much as the sum of each Film Studio's Backpack and each Dance Studio's School Daypack. The number of each Film Studio's School Daypack equals 12 more than the sum of each Film Studio's Messenger Backpack and each Central High's Film Studio. The number of each Central High's Film Studio equals the sum of each Dance Studio's School Daypack and each Film Studio's Messenger Backpack. The number of each Riverview High's Dance Studio equals the sum of each Film Studio's Backpack, each Film Studio's Messenger Backpack, each Film Studio's School Daypack and each Central High's Backpack. The number of each Dance Studio's School Daypack equals 17. The number of each Film Studio's Messenger Backpack equals 13. *How many Backpack does Central High have?*

(2.1)

## 2.2 STEP 2: SOLUTION CONSTRUCTION (CoT)

Let *solution* be a sequence of sentences describing the *necessary* steps towards solving the given problem, where the sentences follow any topological order — also known as Chain-of-Thought, CoT. For each parameter *necessary* towards answering the final question, we assign to it a random letter among the 52 choices (a..z or A..Z), and use a sentence to describe its computation:[7]

$$\text{Define [param] as X; [intermediate steps]; so X = ...}$$

Throughout this paper, we consider **arithmetics mod** 23 to avoid errors from computation involving large numbers. It is perhaps the easiest to directly see a solution example (corresponding to (2.1)), and a more involved example is in Figure 5:

> **(Solution - Easy)** Define Dance Studio's School Daypack as p; so p = 17. Define Film Studio's Messenger Backpack as W; so W = 13. Define Central High's Film Studio as B; so B = p + W = 17 + 13 = 7. Define Film Studio's School Daypack as g; R = W + B = 13 + 7 = 20; so g = 12 + R = 12 + 20 = 9. Define Film Studio's Backpack as w; so w = g + W = 9 + 13 = 22. Define Central High's Backpack as c; so c = B * w = 7 * 22 = 16. *Answer: 16.*

(2.2)

We emphasize that:

- The solution only contain parameters *necessary* towards calculating the final query parameter.
- The solution follows the correct logical order: i.e. all the parameters used in the calculation must have appeared and been computed beforehand.
- We break computations to binary ops: $g = 12 + 13 + 7$ is broken into $g = 12 + R$ and $R = 13 + 7$ in the above solution. The number of semicolons ";" equals the number of *operations*. This reduces the arithmetic complexity of the solution, which is not the focus of this paper.[8]
- Any (abstract) parameter that wasn't mentioned in the problem statement is by default zero.

---

[6]We use simple English sentence templates to describe the problem, and did not worry about grammar mistakes such as singular vs plural forms. There are other sources of randomness besides the dependency graph, such as when parameter $A$ depends on $B, C$ it could be $A + B$ or $A - B$.

[7]There are different ways to format the CoT solution. We noted that starting with "Define [param] as X" instead of [intermediate steps] improves the model's accuracy, so we have adhered to this CoT format.

[8]Even GPT-4 can make mistakes on calculating "3 * (4+10) + 12 * (5+6)" without using external calculator.

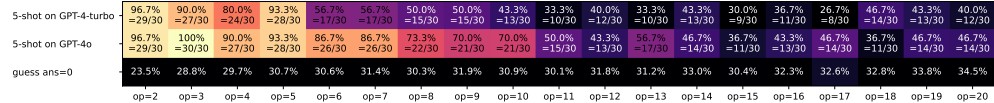

| | op=2 | op=3 | op=4 | op=5 | op=6 | op=7 | op=8 | op=9 | op=10 | op=11 | op=12 | op=13 | op=14 | op=15 | op=16 | op=17 | op=18 | op=19 | op=20 |
|---|---|---|---|---|---|---|---|---|---|---|---|---|---|---|---|---|---|---|---|
| 5-shot on GPT-4-turbo | 96.7% =29/30 | 90.0% =27/30 | 80.0% =24/30 | 93.3% =28/30 | 56.7% =17/30 | 56.7% =17/30 | 50.0% =15/30 | 50.0% =15/30 | 43.3% =13/30 | 33.3% =10/30 | 40.0% =12/30 | 33.3% =10/30 | 43.3% =13/30 | 30.0% =9/30 | 36.7% =11/30 | 26.7% =8/30 | 46.7% =14/30 | 43.3% =13/30 | 40.0% =12/30 |
| 5-shot on GPT-4o | 96.7% =29/30 | 100% =30/30 | 90.0% =27/30 | 93.3% =28/30 | 86.7% =26/30 | 86.7% =26/30 | 73.3% =22/30 | 70.0% =21/30 | 70.0% =21/30 | 50.0% =15/30 | 43.3% =13/30 | 56.7% =17/30 | 46.7% =14/30 | 36.7% =11/30 | 43.3% =13/30 | 46.7% =14/30 | 36.7% =11/30 | 46.7% =14/30 | 46.7% =14/30 |
| guess ans=0 | 23.5% | 28.8% | 29.7% | 30.7% | 30.6% | 31.4% | 30.3% | 31.9% | 30.9% | 30.1% | 31.8% | 31.2% | 33.0% | 30.4% | 32.3% | 32.6% | 32.8% | 33.8% | 34.5% |

Figure 2: GPT-4 (OpenAI, 2023) few-shot accuracies on iGSM-med$_{pq}$ (with $\mathrm{mod}\,5$ arithmetics). For each op we tested 30 problems; and guessing $ans = 0 \in \{0, 1, 2, 3, 4\}$ gives a baseline accuracy around 32%. Details are in the full paper, where we also showcase how GPT-4/4o make mistakes.

## 2.3 DIFFICULTY CONTROL

Although deferring all the data-generation pseudocode to the full paper, we summarize below the main randomness used in the data generation process. This includes the random choice of a hierarchical categorization (i.e., the English part); a structure graph (i.e., the instance parameters); a dependency graph; arithmetic computations on the dependency graph; integer numbers (i.e., the RNG); problem sentence permutation; and the query parameter.

We use two parameters to control data's difficulty: ip is the number of instance parameters, and op is the number of solution operations; the data's difficulty is an increasing function over them. We call our dataset iGSM, to reflect the nature that such synthetic dataset can be of *infinite size*. We use iGSM$^{\text{op}\leq op,\text{ip}\leq ip}$ to denote the data generated with constraint op $\leq op$ and ip $\leq ip$, and use iGSM$^{\text{op}= op,\text{ip}\leq ip}$ to denote those restricting to op $= op$. [9]

## 2.4 TRAIN AND TEST DATASETS

We consider two families of datasets.

- In the iGSM-med data family we use ip $\leq 20$.

  The training data is iGSM-med$^{\text{op}\leq 15}$ $\overset{\text{def}}{=}$ iGSM$^{\text{op}\leq 15,\text{ip}\leq 20}$. We evaluate the pretrained model both in-distribution, on iGSM-med$^{\text{op}\leq 15}$ and iGSM-med$^{\text{op}=15}$, and out-of-distribution (OOD), on iGSM-med$^{\text{op}=op}$ for $op \in \{20, 21, 22, 23\}$ and iGSM-med$^{\text{op}=op,\text{reask}}$. Here, reask denotes first generating a problem from iGSM-med$^{\text{op}=op}$ and then resampling a query parameter.[10]

- In the iGSM-hard data family we use ip $\leq 28$.

  The training data is iGSM-hard$^{\text{op}\leq 21}$ $\overset{\text{def}}{=}$ iGSM$^{\text{op}\leq 21,\text{ip}\leq 28}$. We evaluate the pretrained model both in-distribution, on iGSM-hard$^{\text{op}\leq 21}$ and iGSM-hard$^{\text{op}=21}$, and OOD on iGSM-hard$^{\text{op}=op}$ for $op \in \{28, 29, 30, 31, 32\}$ and iGSM-hard$^{\text{op}=op,\text{reask}}$.

Additionally, we use iGSM-med$_{pq}$ to indicate placing the question *after* the problem and iGSM-med$_{qp}$ the other way (similarly for iGSM-hard). The difficulty of iGSM-med is already quite non-trivial to humans (at least not solvable with few-shot learning using GPT-4/4o, see Figure 2).

**Proposition 2.2.** *Ignoring unused parameters, numerics, sentence orderings, English words, a-z and A-Z letter choices, iGSM-med$^{op=15}$ still has at least 7 billion* solution templates*, and iGSM-hard$^{op=21}$ has at least 90 trillion* solution templates.[11]

---

[9]We choose op non-uniformly; for instance, we let op $= \min\{t_0, t_1\}$ for two random draws $t_0, t_1 \in [op]$. This ensures that the dataset has more easy data — which makes training faster. (See also similar behavior for arithmetics (Jelassi et al., 2023).)

[10]Due to the topological nature of our data/solution generation process, reask greatly changes the data distribution and the number of operations needed. It provides an excellent OOD sample for evaluation. Details are in the full paper.

[11]A solution template is created by replacing all numbers with '0', substituting variables (a-z or A-Z) with letters in their appearance order, and changing parameters to their types (instance or abstract). For instance, "Define Owl Forest's Elephant as y; so y = 11. Define Parrot Paradise's Raccoon as t; so t = y = 11." becomes "Define Inst as a; so a = 0. Define Inst as b; so b = a = 0." We use birthday paradox to estimate the number of solution templates. If $M$ randomly generated problems yield distinct templates, it suggests with good probability that the total number of templates exceeds $\Omega(M^2)$.

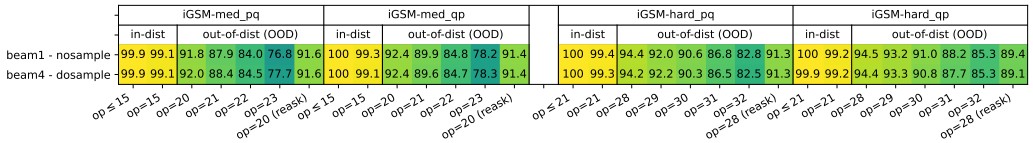

Figure 3: Test accuracies on the model (pre-)trained from the iGSM-med$_{pq/qp}$ and iGSM-hard$_{pq/qp}$ datasets.

| | iGSM-med_pq | | | | | | iGSM-med_qp | | | | | | iGSM-hard_pq | | | | | | | iGSM-hard_qp | | | | | | |
|---|---|---|---|---|---|---|---|---|---|---|---|---|---|---|---|---|---|---|---|---|---|---|---|---|---|---|
| | op≤15 | op=15 | op=20 | op=21 | op=22 | op=23 | op≤15 | op=15 | op=20 | op=21 | op=22 | op=23 | op≤21 | op=21 | op=28 | op=29 | op=30 | op=31 | op=32 | op≤21 | op=21 | op=28 | op=29 | op=30 | op=31 | op=32 |
| avg unnecessary operation | 0.00 | 0.00 | 0.00 | 0.00 | 0.00 | 0.00 | 0.00 | 0.00 | 0.00 | 0.00 | 0.00 | 0.00 | 0.01 | 0.00 | 0.00 | 0.00 | 0.00 | 0.00 | 0.00 | 0.00 | 0.00 | 0.00 | 0.00 | 0.00 | 0.00 | 0.00 |
| avg unnecessary operation (reask) | 0.02 | 0.11 | 0.15 | 0.17 | 0.19 | 0.17 | 0.03 | 0.11 | 0.17 | 0.21 | 0.19 | 0.20 | 0.07 | 0.46 | 0.52 | 0.54 | 0.57 | 0.66 | 0.66 | 0.09 | 0.40 | 0.45 | 0.45 | 0.58 | 0.53 | 0.59 |
| avg unnecessary parameter | 0.00 | 0.00 | 0.00 | 0.00 | 0.00 | 0.00 | 0.00 | 0.00 | 0.00 | 0.00 | 0.00 | 0.00 | 0.01 | 0.00 | 0.00 | 0.00 | 0.00 | 0.00 | 0.00 | 0.00 | 0.00 | 0.00 | 0.00 | 0.00 | 0.00 | 0.00 |
| avg unnecessary parameter (reask) | 0.01 | 0.09 | 0.12 | 0.12 | 0.14 | 0.12 | 0.03 | 0.10 | 0.15 | 0.16 | 0.15 | 0.16 | 0.06 | 0.34 | 0.36 | 0.36 | 0.38 | 0.43 | 0.44 | 0.07 | 0.30 | 0.32 | 0.31 | 0.40 | 0.36 | 0.42 |

Figure 4: Number of unnecessary params / operations used per generated correct solution. Details in full paper.

**No data contamination.** A goal in synthetic math data generation is to prevent data contamination in internet-based math datasets, as noted in Zhang et al. (2024). While it *may be impossible to certify that models trained on internet data are free from contamination*, in our setting, **we can certify this**:

1. We perform OOD evaluation such as on op $\geq$ 28 while providing only op $\leq$ 21 training samples.

2. We train with data whose hash value of *solution template* (see Footnote 11) is $< 17 \pmod{23}$, and test with those $\geq 17$. This ensures *no template-level overlap between training and testing*.

## 3  RESULT 2-3: SUMMARIZE MODEL'S BEHAVIOR PROCESS

We use the GPT2 architecture (Radford et al., 2019) but replacing its absolute positional embedding with rotary embedding (Su et al., 2021; Black et al., 2022), yet still referring to it as GPT2 for short.[12] We mostly stick to the 12-layer, 12-head, 768-dim GPT2 (a.k.a. GPT2-small) for experiments, but we explore larger models in Results 7-8. We use a context length of 768 / 1024 for pretraining on iGSM-med/iGSM-hard and 2048 for evaluation. More details are in the full paper.

**Result 2: accuracy.** After sufficient pre-training, we give the model a problem from the test set (without solution) and let it continue to generate (allegedly a solution followed by an answer). Because we have restricted ourselves to a fixed solution format, language models can learn the format easily, allowing us to write a *solution parser* to check if the solution is fully correct.[13]

Figure 3 shows that GPT2 performs well when pretrained using iGSM-med or iGSM-hard data, even when evaluated out-of-distribution on harder (i.e., larger op) math problems. Thus, the model can truly learn some reasoning skill instead of memorizing solution templates.[14] This could be reminiscent of language models' length generalization capability on arithmetics (Zhou et al., 2023; Jelassi et al., 2023); however, in our case, op captures the "reasoning length" in grade-school math, and our model **has never seen *any*** training example of the same reasoning length as in test time.[15]

---

[12]We also tested with Llama architecture (esp. with gated MLP layers) and did not see major change. GPT2-rotary performs no worse than Llama/Mistral for knowledge tasks (Allen-Zhu & Li, 2025b). We are bounded by resources to repeat all experiments in this paper with other architectures that have small differences from GPT2-rotary.

[13]We check not only the correctness of the final answer 0..22 but also the calculations and parameter dependencies. Language models can learn very complex syntactics, see (Allen-Zhu & Li, 2023) and the references therein.

[14]Llama (of the same model size) gives similar performance, but we refrain from repeating all the experiments with another model. We are not interested in small model differences in this theoretical study; instead, we care more about the general behavior of (autoregressive) language models.

[15]Some others such as Anil et al. (2022) start with a transformer pre-trained on internet data; while the transformer may not have seen the same task during training, it's possible that the model has seen other tasks with the same (or even longer) length and learned to transfer from there.

Such accuracies also indicate that our iGSM data families are indeed good for pretraining purpose, allowing us to investigate further *how* LLMs can solve grade-school math problems.

*Remark* 3.1. Our controlled experiment distinguishes between "reasoning length generalization" and "token length generalization". When designing our test data, we ensured that the test data have a similar token length compared to the training data (though with longer "reasoning length", see full paper for details). Thus, Figure 3 primarily addresses the model's "reasoning length generalization". For readers interested in "token length generalization", we include this also in the full paper.

**Result 3: solution redundancy.**   We examine whether GPT2 achieves high accuracy by

- brute-forcedly computing all the parameters during generation (a "level-0" reasoning skill), or
- computing only necessary parameters to give shortest solutions (a "level-1" reasoning skill).

Recall our iGSM (pretrain) data only contains necessary solution steps (i.e., CoT) to simulate what we see in textbook solutions for math problems. For instance, if a problem describes X=3+2, E=3+X, Y=X+2 and asks for the value of Y, then a shortest solution would be "X=3+2=5 and Y=X+2=7" without ever computing E.

Figure 4 shows that GPT2 predominantly solves the iGSM problems with a "level-1" reasoning skill, avoiding unnecessary computations, even when evaluated out-of-distribution. This finding is significant as it suggests that, unlike humans who usually rely on "backward reasoning" and a scratch pad to write down necessary parameters by backtracking the dependencies from the question (Rips, 1994), the language model can directly generate shortest solutions without using a scratch pad. But, how does it achieve so? We shall investigate in the next section.

## 4   RESULTS 4-8?

Due to space limitation, we omit the technical details for Results 4-8 in this ICLR version to encourage readers to refer to our full paper at `ssrn.com/abstract=5250629`. We remark that the full paper underwent the ICLR 2025 review process, but we elected to present this camera-ready version as an *extended abstract*, aligning with the tradition in the theory community.

## 5   CONCLUSION

We use a synthetic setting to demonstrate that language models can learn to solve grade-school math problems through true generalization, rather than relying on data contamination or template memorization. We develop probing techniques to examine the models' hidden reasoning processes. Our findings reveal that these models can learn math skills aligned with human cognitive processes, as well as "new thinking processes" not present in the training data. Additionally, we propose a method to predict a model's errors before it begins to solve a problem and to explain why models make mistakes when they occur. Based on this discovery, we write a separate paper to improve language models' math reasoning accuracy (Ye et al., 2025). We also provide a principled approach to connect the model's depth to its capable reasoning length. We believe this research opens doors to study the mathematical reasoning skills of language models from a different angle compared to pushing math benchmarks.

One may argue that iGSM may be very different from the pretrain data that modern LLMs use. While this may be true, we are looking into the future. Recall, even GPT-4/4o of today cannot few-shot learn to solve iGSM-med$^{\text{op}=11}$ (see Figure 2). From this perspective, it is reasonable to believe that future versions of LLMs will rely on synthetic math pretrain data to improve their reasoning skills. While one may not directly use iGSM, it is tempting to use existing LLMs (such as Llama-3) to turn iGSM into more natural formats while keeping the logical chains. On the other hand, we have discovered that models trained purely on the iGSM data make similar mistakes compared to GPT-4/4o (see full paper). This further confirms that our findings do connect to practice, regarding the model's hidden reasoning process.

Finally, Part 2 of this work series focuses on how language models solve grade-school math problems (including Part 2.2 (Ye et al., 2025)). We also cover how language models learn language

structures in Part 1 (Allen-Zhu & Li, 2023) (in particular, how they mentally perform dynamical programming), learn world knowledge in Part 3 (Allen-Zhu & Li, 2024; 2025a;b), and how these may impact architecture design (Allen-Zhu, 2025).

## 6 AN EXAMPLE IN iGSM-HARD WITH OP = 21

**(Problem - A Hard Example)** The number of each Jungle Jim's International Market's Cheese equals the sum of each Parmesan Cheese's Pear and each The Fresh Market's Ice Cream. The number of each Ice Cream's Pineapple equals 2 more than each Goat Cheese's Grape. The number of each New Seasons Market's Goat Cheese equals the sum of each Residential College District's Jungle Jim's International Market, each Jungle Jim's International Market's Parmesan Cheese and each Residential College District's Supermarket. The number of each Arts Campus's New Seasons Market equals each Cheese's Pineapple. The number of each Goat Cheese's Banana equals each Vocational School District's Product. The number of each Residential College District's Jungle Jim's International Market equals 5 more than each Ice Cream's Grape. The number of each Parmesan Cheese's Pineapple equals each Parmesan Cheese's Pear. The number of each Residential College District's The Fresh Market equals each Arts Campus's Trader Joe's. The number of each Arts Campus's Trader Joe's equals each Parmesan Cheese's Ingredient. The number of each Goat Cheese's Grape equals 0. The number of each The Fresh Market's Ice Cream equals 13 more than the difference of each Residential College District's The Fresh Market and each Parmesan Cheese's Grape. The number of each Goat Cheese's Pineapple equals New Seasons Market's Product. The number of each Vocational School District's The Fresh Market equals the sum of each Trader Joe's's Cheese and each The Fresh Market's Cheese. The number of each Trader Joe's's Cheese equals 6. The number of each The Fresh Market's Cheese equals 3. The number of each Jungle Jim's International Market's Ice Cream equals the difference of each Ice Cream's Banana and each Goat Cheese's Grape. The number of each Jungle Jim's International Market's Parmesan Cheese equals each Ice Cream's Pineapple. The number of each Parmesan Cheese's Pear equals the difference of each Goat Cheese's Grape and each Ice Cream's Grape. The number of each Parmesan Cheese's Grape equals 12 times as much as each Residential College District's Jungle Jim's International Market. The number of each The Fresh Market's Parmesan Cheese equals each The Fresh Market's Cheese. The number of each Ice Cream's Banana equals the sum of each Parmesan Cheese's Pineapple and each Ice Cream's Pineapple. The number of each School District's Jungle Jim's International Market equals each The Fresh Market's Ice Cream. The number of each Cheese's Pineapple equals 20 more than the sum of each Trader Joe's's Cheese and each The Fresh Market's Cheese. The number of each Trader Joe's's Parmesan Cheese equals 16. The number of each Ice Cream's Pear equals 8. The number of each Ice Cream's Grape equals each Goat Cheese's Grape. *How many Product does School District have?*

**(Solution - A Hard Example)** Define Goat Cheese's Grape as u; so u = 0. Define Ice Cream's Grape as x; so x = u = 0. Define Residential College District's Jungle Jim's International Market as N; so N = 5 + x = 5 + 0 = 5. Define Parmesan Cheese's Pear as G; so G = u - x = 0 - 0 = 0. Define Parmesan Cheese's Grape as f; so f = 12 * N = 12 * 5 = 14. Define Parmesan Cheese's Pineapple as C; so C = G = 0. Define Parmesan Cheese's Ingredient as Z; e = f + C = 14 + 0 = 14; so Z = e + G = 14 + 0 = 14. Define Arts Campus's Trader Joe's as q; so q = Z = 14. Define Residential College District's The Fresh Market as j; so j = q = 14. Define Ice Cream's Pineapple as X; so X = 2 + u = 2 + 0 = 2. Define Ice Cream's Banana as K; so K = C + X = 0 + 2 = 2. Define The Fresh Market's Ice Cream as P; i = j - f = 14 - 14 = 0; so P = 13 + i = 13 + 0 = 13. Define Jungle Jim's International Market's Ice Cream as R; so R = K - u = 2 - 0 = 2. Define School District's Jungle Jim's International Market as V; so V = P = 13. Define Jungle Jim's International Market's Cheese as v; so v = G + P = 0 + 13 = 13. Define Jungle Jim's International Market's Parmesan Cheese as S; so S = X = 2. Define Jungle Jim's International Market's Product as y; U = S + R = 2 + 2 = 4; so y = U + v = 4 + 13 = 17. Define School District's Product as J; so J = V * y = 13 * 17 = 14. *Answer: 14.*

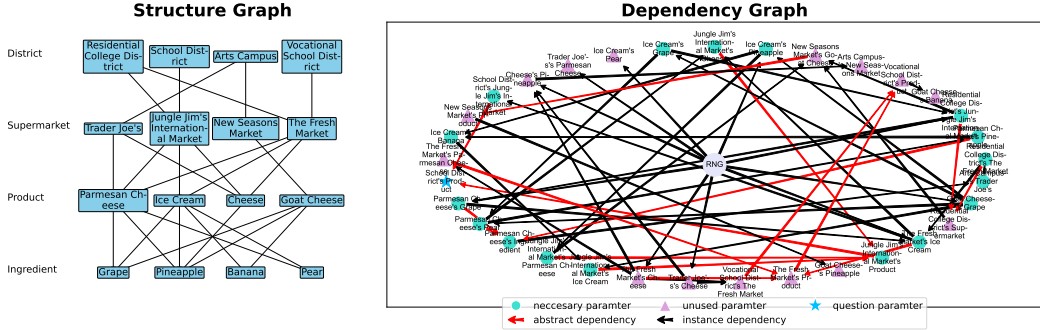

Figure 5: An example with op = 21 in iGSM-hard$_{pq}$ used for training. Don't forget during testing we evaluate models on op = 32 which is even much harder.

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
