# OpenReview forum: "Physics of Language Models: Part 2.1, Grade-School Math and the Hidden Reasoning Process"
_ICLR.cc/2025/Conference — ICLR 2025 Poster_

### Official Review · Reviewer_35Ln · 2024-10-27

**Soundness:** 2
**Presentation:** 2
**Contribution:** 2
**Rating:** 6
**Confidence:** 4

**Summary:**

This work focuses on evaluating the grade-school level mathematical reasoning of neural language models (GPT models).
The authors set out to explore several critical questions surrounding the actual abilities of LMs to reason mathematically.
To make clear and controllable experiments, the authors introduces a synthetic dataset iGSM.
Using iGSM to train and test GPT-2, the authors argue that the model can truly understand and reason through problems rather than just memorizing instances.
The examination of the causes behind reasoning mistakes made by LMs is another highlight, shedding light on the limitations and potential areas for improvement.

**Strengths:**

1. Overall, the exploration of the model's hidden reasoning process provides valuable insights into the black-box nature of neural language models.
2. This work introduces a new synthetic dataset iGSM which could facilitate the training and evaluation of contemporary large language models.
3. This work provides many interesting findings. It provides a valuable addition to better understanding the reasoning capabilities emerged in large language models.

**Weaknesses:**

My major concern is about the generalizability of the conclusions drawn in this work.
It is unclear whether the experimental findings on formal language reasoning can be generalized to natural language reasoning.
The formal language patterns in iGSM are too much limited and inflexible compared with natural language expressions, although it may contain a large number of templates if omitting the primitives in the expressions.

For instance, for the math operation *multiplication*, there are many different ways to express using natural language, e.g., "there are three children in the house and each child eats two eggs for breakfast".
But in iGSM, it seems that *multiplication* is always mapped from "times" in the input expression.
I think for grade-school level mathematical reasoning, one major capability is to map the flexible natural expressions to calculations.
Therefore, I think reasoning on iGSM could be much easier than reasoning on natural language questions.

**Questions:**

1. Can you provide some evidence to support the generalizability of your conclusions, even some of them?

---

> ### Author Response · Authors · 2024-11-30
> **Response to Reviewer 35Ln [1/2]:**
>
> We sincerely thank Reviewer 35Ln for raising the generalizability concern. Quite conversely, we find it more beneficial to exclude for instance different ways to express “multiplication” in order to perform more controlled experiments. Let us explain.
>
> Our paper is very unconventional (and innovative) in the sense that we have deviated from most of the work that uses real-life data to directly improve model’s benchmark accuracies. We wish to focus on one specific aspect of LLM and try to push it to a limit, and study issues in this aspect, possibly predicting what future LLMs need.
>
> With this said, there are many different aspects of “intelligence” and we wish to focus on the logic-following part by removing most others.
>
> * “Arithmetics” is one such aspect. For instance, LLMs can be better at arithmetics when the digits are reversed (especially when digits are single-tokened) and there’s rich literature studying this.
>
> * “Common sense or factual knowledge” is another such aspect, ranging from “Paris is the capital of France” to “theraphosa blondi (a type of spider) has 4 legs”. One may need a very large model to capture all factual knowledge, and there are many LLM issues such as reversal curse, etc., see prior work 2309.14402.
>
> * “Translation” is an aspect that relates to your mention of the “multiplication” example. This is commonly known as the difficulty of the “math word problem”, where the student needs to learn to associate phrases such as "surplus, more, total, together" to the addition symbol. This is similar to factual knowledge, so it’s one thing that we have **intentionally avoided** when building this dataset, because we want to focus on the “logic-following” issues with long reasoning chains.
>
> * Please note even with all the aspects above are removed, GPT-4o still completely fail on such "logic-following" data for op>=11, and largely fail for op>=8 (see Fig 2 + 17); and a follow up (2408.16293) verified that GPT-4o fails even for much simpler variants of iGSM (namely, "Box A contains a Box B and it weights 7lb on its own.") That's **precisely why it is worth focusing on this logic following part alone** to understand a transformer's limit and inner working.  If one instead studies all of the issues together, especially in a black-box manner, it can be misleading --- see the recent debate on the Reflection model (https://huggingface.co/mattshumer/Reflection-Llama-3.1-70B), where there seems to be an improvement on the reasoning benchmarks, but actually...
>
> > I think reasoning on iGSM could be much easier than reasoning on natural language questions.
>
> We particularly want to debate on this statement. While it can be challenging to define “easy”, we kindly ask the reviewer to try to estimate your (human) time needed to solve an average GSM8k problem vs an iGSM problem with op>=20. We bet even if one is well-trained on iGSM problems, humans may need >=5min to figure out the answer on iGSM but much shorter on GSM8k. The difficulty arises on the topological sort part, not the English translation (from “times” to “x” symbol).
>
> Again, we agree that GSM8k has many other “math word problem” difficulties, such as when candle burns its length shrinks (not increases). This is more related to factual knowledge or “translation” as we mentioned above, and is what we have explicitly avoided in this study.

---

> ### Author Response · Authors · 2024-12-01
> **Response to Reviewer 35Ln [2/2]**
>
> **Question:**
>
> > Can you provide some evidence to support the generalizability of your conclusions, even some of them?
>
> **Our response:**
> We response this by carefully delving into what "generalizability" means in this context.
>
> To us, it means for this (and similar) logic-following type of reasoning, as long as training data is sufficient and of high quality, and model is sufficiently deep, then transformer is **capable of learning** to perform such logic following, and the inner working **follows our probing**.
>
> If by "generalizability" you mean whether the finding is generalizable to LLMs trained on internet data, then as we said, today’s LLMs like GPT-4o largely fail on iGSM even with op>=11, so **it is not interesting to probe GPT-4o** (even if we can) unless perhaps op<=7, but that’s too simple a logic-following problem. (Likewise, the logic in problems in GSM8k are too simple and not worth probing.) On the other hand, we verified that the types of errors that GPT-4o fail (see Fig 17) exactly coincide with how our pretrained GPT-2 fail, so this is some kind of generalizability.
>
> If by "generalizability" you mean whether our data can be used to train better all-purpose LLMs, then probably _not directly_. This paper tries to predict the future (after internet data runs out) and studies the limit of the transformer and what’s the data/CoT format needed to achieve such performance, etc. While for today’s models, it appears easy to make improvements --- say can use GPT-4 generated problems to train Phi-3 --- but this will reach a bottleneck; otherwise GPT-4 can improve to GPT-5 already. To overcome this, one had better understand more inner workings (e.g., length of reasoning in internal states) about LLMs to do things **less blindly and more scientifically**. This can teach us for instance what are the necessary data formats to support this (such as “backward thinking” is not needed as part of CoT, Line 447-453), and what type of reasoning mistakes a model could still make (Section 5 and Appendix A).
>
> With this in mind, one wishes to find out what type of program-generated synthetic data can improve LLM's reasoning (after internet data runs out). While our iGSM looks a lot simpler than IMO, powerful models like GPT-4o still largely fail on it -- so iGSM is actually one such starting example of program-generated data, applicable even for the state of the art models.
>
> If by "generalizability" you mean whether our methodology is generalizable to other tasks, then it depends on what "others" mean.
> * If it's any logic-following / logic induction type of reasoning then it does generalize --- just a formatting/translation issue and LLMs are superb at learning formats --- that's why they hallucinate when they don't know the answer, because they have learned the format.
> * If it's logic-following reasoning PLUS other aspects, such as common-sense knowledge or arithmetics as we discussed in [1/2], then all other issues need to be taken into account altogether, and the learning difficulty will be a combination of all of them. As we pointed out, we wish to focus in this paper so we have removed them.
> * If it's some more complex forms of reasoning — such as those needed to prove a hard math problem, we are not there yet. We believe that requires more controlled experiments to scientifically study how LLMs can best do “planning” (such as a DFS search). We generally wish to start from the simplest to understand how LLMs function, and hope to start from here. What do you think of this plan?
>
> Thanks for your time reading our very detailed reply!

---

> ### Comment · Reviewer_35Ln · 2024-12-03
> **Response to Authors**
>
> Thanks for the authors response. The detailed discussion on the generalizability has partially adressed my concerns.
> I have raised the score to 6.

---

### Official Review · Reviewer_Arcv · 2024-11-03

**Soundness:** 3
**Presentation:** 3
**Contribution:** 2
**Rating:** 6
**Confidence:** 5

**Summary:**

The paper investigates the capability of language models (the authors used the GPT-2 model for their experiments), to solve grade-school math problems to explore whether models genuinely develop reasoning skills beyond template memorization. They address several key questions: Can LMs generalize reasoning skills? Do they replicate human-like problem-solving methods? Are their errors predictable, and does model architecture (depth versus width) influence reasoning abilities?

It is important to control the dataset to explore these questions. For this, the authors generate a synthetic dataset with high template diversity, reducing contamination and ensuring complex, logical dependencies in problems. Results indicate that the model can indeed generalize reasoning processes and even learns dependencies and plans solutions. The paper explores the models internal states through probing to understand how the model is solving the problem.

Overall, the paper is highly experimental with the goal to identify the factors that enable a model to develop reasoning skills through systematically controlled experiments.

**Strengths:**

The biggest strength of the paper is the controlled experiments that the authors conducted. Training a GPT-2 model from scratch with their own synthetic data can give them full control over the experiments. Also, generating their own data can prevent contamination of the training test data, which might be one of the problems in the other benchmark data.

The authors also introduce the probing tasks to explore the internal states of the model, which gives some idea of the reasoning process of the model. In particular, the V-probing approach is interesting because it can tell whether pre-training or fine-tuning was responsible for the probing signal.

Finally, the paper presents many interesting results, including the depth vs. width of the models, the generalizability of the model to unknown tasks of greater depth, models learning backward reasoning processes like humans, and so on. These results (if they can be scaled) may allow better design of language models in the future and may help to understand the role of synthetic data in learning a concept.

**Weaknesses:**

The weakness of the paper also lies in the main choice of experiments: the creation of the dataset and the choice of the model.
- Regarding the dataset, the use of synthetic data may not fully capture the complexity and nuances of real-world mathematical problems. This limitation could affect the generalizability of the results to practical applications. It would be important to mix and match the training dataset with real-world data or test the methodology on some real-world data such as GSM8K.
- Regarding the models, although it is difficult to test the methodology on multiple models, the model performance scales well with depth is kind of a known fact and with more layers, the performance improves creating a question that if scaling leads to memorization?

The use of math word problems is a good choice to control the experiments, but the question is if the results are limited to a structured reasoning problem like math word problems? This is important to know in order to understand the limitations of the approach.

The analysis of the model errors was omitted from the paper, saying that the authors are writing another paper because it is not good for the current paper. It would be good to see the error analysis. At least a summarized version needs to be presented.

Finally, since the training of the model was done by the authors, it would be interesting to see in which training iterations the model learned certain skills. This can easily be discussed in the paper, but I found it missing.

[MINOR] The use of words like AGI while experimenting with synthetic datasets and the GPT-2 model is not correct and can be avoided.

**Questions:**

It would be interesting to see some results on real-world datasets like GSM8K to see how much the models can generalize to real-world problems when trained on synthetic datasets.

Is it possible to test the models on similar synthetic datasets like mathworld (https://arxiv.org/abs/2306.04347)?

Model errors and how LLMs can fix them (Result 6) needs to be discussed in the paper and not left saying its there in another paper.

The results of the backward thinking process are missing from the paper and saying that the models learned it without any discussion is not correct. It needs to be added to the paper.

Finally, it would be interesting to see an analysis of when a model learned a particular reasoning skill during training. And can it be learned in a continuous training fashion?

---

> ### Author Response · Authors · 2024-11-30
> **Response to Reviewer Arcv [1/3]**
>
> We would first like to sincerely thank the reviewer for raising so many detailed questions.  Your time on reviewing this paper is much appreciated! Let us try to address your questions one by one.
>
> **Question:**
> > Regarding the dataset, the use of synthetic data may not fully capture the complexity and nuances of real-world mathematical problems. This limitation could affect the generalizability of the results to practical applications. It would be important to mix and match the training dataset with real-world data or test the methodology on some real-world data such as GSM8K.
>
> **Our response:**
> First of all, we hope to point out this paper is unconventional compared to most LLM works these days. We are not talking about improving benchmarks, so a first question is whether our **probing results** apply to real-life models?
>
> For this question, the answer is No if you’re concerning models of today but will be Yes for models of tomorrow. Today’s models such as GPT-4o and Llama3 can still fail quite badly on iGSM data (complete failure on op>=11) or even its much simpler data variants (see follow ups such as 2408.16293). Therefore you won’t expect good probing results unless perhaps op<=7, but that’s too simple a problem not worth probing. (Likewise, the logics in problems in GSM8k are too simple and not worth probing.)
>
> Furthermore, another main goal of this paper is to make suggestions on what’s the data/CoT format needed to train tomorrows' next-generation LLMs. **We are not concerned about commercial models of today** --- to improve most models today it can still be easy, just use GPT-4o to generate more data and you can improve reasoning benchmarks. But this will eventually reach a bottleneck; otherwise GPT-4 can improve to GPT-5 already. To overcome it, one had better understand more inner workings about LLMs (e.g., length of the reasoning chain in internal states) to do things **less blindly and more scientifically**. This can teach us for instance what are the necessary data formats (such as “backward thinking” is not needed as part of CoT, which we answer your question later), and what type of reasoning mistakes a model could still make (again, we respond in more detail later).
>
> As for the **the complexity and nuances of real-world mathematical problems**, real-life math problems have many different aspects, including (A) common-sense knowledge and (B) arithmetics. When combined, then all the issues related to (A) and (B) must also be taken into account. For instance, (A) has issues such as the knowledge manipulations (see 2309.14402) and (B) has issues that one needs to reverse the digit orders for better arithmetics. There’s rich literature to study each individual problem (in a synthetic manner) and we expect that we need to gather all the findings from individual studies to eventually help build the next generation of LLMs. If one tries to study all of the issues together, especially in a black-box manner, it can be misleading --- see the recent debate on the Reflection model (https://huggingface.co/mattshumer/Reflection-Llama-3.1-70B), where there seems to be an improvement on the reasoning benchmarks, but actually...
>
> Moreover, there're more complex forms of reasoning — such as those needed to prove a hard math problem --- and we have not covered yet. We believe that requires a few more steps (likely also with synthetic controlled experiments) to scientifically study how LLMs can best do “planning” (such as a DFS search, in which the context length issue will also kick in). We generally wish to start from the simplest to understand how LLMs function, and hope to start from here. What do you think of this plan?
>
>
>
>
>
> **Question:**
> > Regarding the models, although it is difficult to test the methodology on multiple models, the model performance scales well with depth is kind of a known fact and with more layers, the performance improves creating a question that if scaling leads to memorization?
>
> **Our response:**
> Not necessarily. For factual knowledge, increasing model depth (while keeping model size unchanged) does not improve performance, see 2404.05405. It is most important that we are doing this in a controlled setting, by ensuring that the data / model sizes are controlled, and we see that deeper models are especially better at longer reasoning chains.
>
> We don’t think this is related to memorization. As we pointed out, even if you remove all the English names and arithmetic numbers, the description of the solution (what we call “templates”) still has 90 trillion possibilities, see Line 70.

---

> ### Author Response · Authors · 2024-11-30
> **Response to Reviewer Arcv [2/3]**
>
> **Question:**
> > The use of math word problems is a good choice to control the experiments, but the question is if the results are limited to a structured reasoning problem like math word problems? This is important to know in order to understand the limitations of the approach.
>
> **Our response:**
> We refrain from viewing our studied problem as “math word problem”, because in most math word problems, the difficulty is more linked to the “translation” from the real-life description to the math. For instance, when seeing "surplus, more, total, together" the grade school student is supposed to learn to use "addition", or more difficult, when “a candle burns” its length shrinks instead of increases. Some of these are more connected to translation / formatting, and some are more connected to world knowledge. We have tried to remove both from our study, because we want to focus more on the “logic following” for longer reasoning chains. As you may see, our reasoning is typically of much longer solution steps compared to GSM8k.
>
>
>
> **Question:**
> > The analysis of the model errors was omitted from the paper, saying that the authors are writing another paper because it is not good for the current paper. It would be good to see the error analysis. At least a summarized version needs to be presented.
>
> **Our response:**
> Indeed due to space limitation we unfortunately have to defer the model errors to the appendix. The quick summary is that, model makes two types of mistakes on this iGSM data:
> (1). One is that the model occasionally computes unnecessary parameters (for very hard problems, such as op>=21). We used probing to actually discover this links to the model’s mental process stage. That is, the model actually did the topological sort wrong in its preprocessing, before ANY token is generated.
> (2). The other is that the model sometimes defines a parameter that is not ready for computation, and then will start to make mistakes. We again probed and discovered that the model may mentally indeed *initially* think that the parameter is computable, but while it starts the calculation, it then figures out this is not possible and start to hallucinate (meaning to write down in the correct format but a wrong calculation).
>
> It’s worth noting not only our trained GPT-2 makes these two types of mistakes, also GPT-4o on our synthetic data make these two types of mistakes.
>
> All the results above are included in this paper in the appendix (just one page, in Appendix A). We can move things around to include more of it upon your request. It’s worth noting that we have written a separate paper to study how to fix issue (2). The main idea is to introduce a <BACK> symbol to allow the model to backtrack, but with many more findings that cannot fit into this paper.
>
> **Question:**
> > Finally, since the training of the model was done by the authors, it would be interesting to see in which training iterations the model learned certain skills. This can easily be discussed in the paper, but I found it missing.
>
>
> **Our response:**
> That’s a great question and let us respond as briefly as we can. Most skills are not learned in a grokking fashion, in the sense that during the course of the pretraining, the test accuracies increase faster for small op, then for medium op, and finally for large op. Grokking, on the other hand, means that some skill is suddenly learned at some point of the training.
>
> We can supplement an experiment explaining this, but we are not sure how interesting it is.
>
> Only in one place we discovered grokking. Namely, remember our objects are in four category layers (e.g., School vs Classroom vs Backpack vs Stationery). While it is easy to compute “the number of Classrooms for a given high school”, It becomes increasingly hard to go across category layers. The model needs to learn to do all the necessary pre-computations before it can calculate the number of stationeries in a high school (say, the high school has 4 types of classrooms, each having some backpacks, each having rulers and erasers, etc). The model needs some training steps to learn to perform such computations, and this is the only place where we have observed the grokking effect. Unfortunately, due to the space limitation, and since this is somewhat out of the scope of this paper, we have decided not to include it.

---

> ### Author Response · Authors · 2024-11-30
> **Response to Reviewer Arcv [3/3]**
>
> **Question:**
> > The results of the backward thinking process are missing from the paper and saying that the models learned it without any discussion is not correct. It needs to be added to the paper.
>
> **Our response:**
> Our results are complete and sorry if we have confused you.
>
> Suppose the problem describes some <param1> that is 4 times <param2>, then our CoT of iGSM is formatted as: “.... Define <param2> as X, so X=... Define <param1> as Y, so Y = 4 * X …” Note:
> * we have never explicitly written down (as CoT) that “because <param1> depends on <param2> so we need to first compute <param2>.
> * we also never put any tag on top of <param2> such as "the purpose of this is to compute <param1>"
>
> In sum, we have **not included backward thinking** in the data, yet GPT-2 can still be trained to perform the desired logic **mentally**. Therefore, our conclusion is that, the “backward thinking” for such logic problems is not needed to be present as CoT, as long as the training data is sufficient and when transformers are not too shallow.
>
> In case you’re interested, once you add such backward thinking as CoT, then the mental process becomes much shallower, and the model no longer needs to think deep mentally such as “A depends on B, B depends on C, C depends on D, and I haven’t computed D yet. Let’s next compute D!”
>
>
> Thanks again for your time giving us very detailed feedbacks.

---

> > ### Comment · Reviewer_Arcv · 2024-12-03
> >
> > Thank you for your thorough response.
> >
> > The discussion around generalizability is much clearer to me now, and I appreciate the authors' transparency in outlining their goals. While the paper provides valuable insights, adding an analysis of model errors and further elaboration on the limitations of the approach would enhance its completeness and make it more accessible to readers.
> >
> > I have already assigned a score of 6, which reflects a positive score from my side and I would like to maintain the score.

---

### Official Review · Reviewer_GM4V · 2024-11-04

**Soundness:** 4
**Presentation:** 3
**Contribution:** 3
**Rating:** 6
**Confidence:** 2

**Summary:**

This work conducts controlled experiments to probe several fundamental questions about math reasoning capability of language models, the experimental results show that language models can solve math problems like humans and conduct backward thinking process.

**Strengths:**

- This work introduces a framework to generate a set of diverse GSM problems that focus on “logical reasoning” aspect.
- This work finds that “the model has learned to plan ahead, identifying necessary
parameters before starting to generate the solution”.

**Weaknesses:**

- The small model size of GPT2-small (100M-200M) may effect the generalization of the experimental conclusion. It would be appreciated if conducted experiments on LLMs with billion parameters.
- The conclusions drawn from experiments on synthetic data may not be applicable to real-world scenarios with large data volumes.

**Questions:**

Since IGSM only has four categorizations, which may result in poor diversity in the description of quantitative relationships. Could this affect the generalization of the experimental conclusions of this paper?

---

> ### Author Response · Authors · 2024-11-30
> **Response to Reviewer GM4V [1/2]**
>
> We sincerely thank the reviewer for reading the paper and raising the questions. Let us start with your simplest question.
>
> **Question:**
> > Since IGSM only has four categorizations, which may result in poor diversity in the description of quantitative relationships. Could this affect the generalization of the experimental conclusions of this paper?
>
> **Answer:** First of all, four categorizations each have four layers, and each layer has ~100 object names (totaling ~1600), and some object names may also be ambiguous (just like “backpacks” and “laptop backpacks” mean two different things). So we at least wish to say that this is more complex than using fake object names A-Z.
>
> Of course, real-life situations can be more complex, and the quantitative relationship is a form of “factual knowledge”, and this can range from “Paris is the capital of France”, to “theraphosa blondi (a type of spider) has 4 legs”. This paper tries to separate such common-sense knowledge from reasoning, in order to focus. If instead one has many more such categories, such as 16 million objects, then the model’s failure might be due to its poor learning of such factual knowledge, and not to say “LLM vs. factual knowledge” has many orthogonal issues to reasoning (see prior work 2309.14402). We think this is like studying Newton’s laws in a vacuum chamber, so we want to remove potential noise from other aspects of LLMs.
>
>
> **Question:**
> > The small model size of GPT2-small (100M-200M) may effect the generalization of the experimental conclusion. It would be appreciated if conducted experiments on LLMs with billion parameters.
>
> **Answer:**  This is a valid concern in most LLM studies when real-life data is involved, but we do not believe it applies to our setting. When considering real-life data and benchmarks, and especially when considering the “emergent ability” of LLMs, there’s too much to cover and one needs a model to be >=7B in size.  (Note, if in the future when data quality is improved, one may not need 7B and this threshold could potentially lower down, see the “Junk vs Scaling law” section of prior work 2404.05405).
>
> When studying our synthetic setting, by using models of 100-200M size, we have already managed to solve iGSM problems that GPT-4o is not capable of. Further increasing model size will certainly further improve the hardness of the iGSM problems one can solve, but let us make an analogy. It's like Galileo’s Pisa Tower experiment, where we care more about in a controlled setting how two experiments compare; and when the height of the Pisa tower is tall enough, it may not be too necessary to further increase it. It is of course always better to study larger models, but the cost-benefit tradeoff may become marginal.
>
> Specifically, in the most ideal case, say increasing the model size by 5x, also increases the iGSM difficulty by 2x, resulting in 2x the context length, and possibly 4x training problems. When combining this together, it is going to cost us 40x+ more $$$ to repeat the same experiments. We are short of money to perform such experiments at this moment.

---

> ### Author Response · Authors · 2024-11-30
> **Response to Reviewer GM4V [2/2]**
>
> **Question:**
> > The conclusions drawn from experiments on synthetic data may not be applicable to real-world scenarios with large data volumes.
>
> **Answer:**
> Let us break your question down to a few aspects.
>
> If you’re concerned that our probing results may not apply to real-world scenarios, let us quickly point out it won’t apply to models of today but will apply to models of tomorrow. Today’s models such as GPT-4o and Llama3 can still fail quite badly on iGSM data (completely failure on op>=11) or even its simpler data variants (see followups such as 2408.16293). Therefore you won’t expect good probing results unless op<=7, but that’s too simple a problem not worth probing. In contrast, the main goal of this paper is to test a transformer's limit on such “logic-following” reasoning when data is abundant. We are not concerning any commercial model of today, but study “when data is abundant and of high quality” tomorrow, what’s the limit of the next-generation models.
>
> For other conclusions of this paper, such as whether the “backward thinking CoT” is needed, and the “depth matters”, and how LLMs make mistakes on such reasoning tasks, we do believe they apply to models of tomorrow (and some even to today’s models, such as the reasoning mistakes, as we showed in Figure 17).
>
> If you're concerned whether our data can be used to directly train better all-purpose LLMs, then probably _not directly_. This paper tries to predict the future (after internet data runs out) and studies the limit of the transformer and what’s the data/CoT format needed to achieve such performance, etc. While for today’s models, it appears still easy to make improvements --- say can use GPT-4 generated problems to train Phi-3 --- but this will reach a bottleneck; otherwise GPT-4 can improve to GPT-5 already. To overcome this, one had better understand more inner workings about LLMs to do things **less blindly and more scientifically**. This can teach us for instance what’s the data/CoT format needed, and what's the type of program-generated reasoning data needed, etc. While our iGSM looks a lot simpler than IMO, powerful models like GPT-4o still largely fail on it --- so iGSM is actually one such starting example of program-generated data (with CoT) applicable even for the state of the art models.
>
> This paper is unconventional compared to most other LLM papers in the field, because it can be a "bless" to study synthetic controlled experiments — like studying Newton’s laws in a vacuum chamber, so we can remove the noise (from such as data quality, world knowledge, arithmetics) and focus. If you don't remove the noise and study blindly on reasoning, a bad example is the recent debate on the Reflection model (https://huggingface.co/mattshumer/Reflection-Llama-3.1-70B), where there seems to be an improvement on the reasoning benchmarks, but actually...
>
> Of course, we are only in Step 1 towards understanding “reasoning” of LLMs, and there are many more complex forms of reasoning waiting for us to understand better, such as the planning needed to solve hard math problems. We sincerely believe that preparing synthetic data can also be needed there, for instance, to study how LLMs can better and more cleverly perform “search” in its planning space to reduce reasoning token length, etc.  We generally hope to bring our findings/insights on controlled experiments to real-world applications, but not to directly apply it (such as directly applying iGSM data in real world is not what we recommend).

---

### Official Review · Reviewer_UB35 · 2024-11-05

**Soundness:** 3
**Presentation:** 3
**Contribution:** 3
**Rating:** 6
**Confidence:** 4

**Summary:**

This paper dives deep into how language models solve grade-school math problems. The authors first design an elaborated data generation pipeline to synthesize a large amount of grade-school math problems with a specific form of Chain-of-Thoughts. Then the authors design a probing method to analyse how the language models solve the problems, and acquire some interesting conclusions.

**Strengths:**

* This paper carries out a substantial experiment (including large-scale data synthesis, fine-tuning, and probing) to systematically research the reasoning ability of LLMs in grade-school math problems.
* To understand how the metal process of language models, the paper proposes a few **probing tasks** which align with human problem-solving strategies. Such probing tasks might be insightful to other works.
* Multiple interesting conclusions are found in this paper.

**Weaknesses:**

* **Limitation of specific CoT form**: This paper trains the GPT-2 model using synthetic data with a specific form of CoT. The fune-tuning and probing experimental conclusions are based on such CoT. However, the CoT that is now used to train the LLMs on grade school math problems deviates considerably from this form. This limits the generalizability of the paper's conclusions.
* **Lack of analysis on data quantity**: Although this paper synthesizes a large amount of data to train a language model, some studies have shown that more training data is not always better for language models. I expect to see a curve or table of [model performance - the quantity of training data].

**Questions:**

1. **Lack of analysis on data quantity**: See the second point in Weaknesses.
2. **About the backward thinking process**: In line 447-453 of this paper, the authors define the “backward thinking process” as “because I want to compute X, but X depends on Y and Y depends on Z, so let me compute Z first”. Moreover, they conclude that this backward thinking process can be autonomously learned through language modeling with abundant data. **I can't understand how the author came to this conclusion**. I cannot understand how the authors arrive at this conclusion.

---

> ### Author Response · Authors · 2024-11-30
> **Response to Reviewr UB35 [1/2]**
>
> We first would like to thank the reviewer for appreciating our probing efforts and liking our substantial experiment. Let us try to address your questions.
>
> ## simple question first
>
> > About the backward thinking process: In line 447-453 of this paper, the authors define the “backward thinking process” as “because I want to compute X, but X depends on Y and Y depends on Z, so let me compute Z first”. … I cannot understand how the authors arrive at this conclusion.
>
> **Answer:** sorry for the confusion. Suppose the problem describes some <param1> that is 4 times <param2>, then our CoT of iGSM is formatted as: “.... Define <param2> as X, so X=... Define <param1> as Y, so Y = 4 * X …” Note:
> * we have never explicitly written down (as CoT) that “because <param1> depends on <param2> so we need to first compute <param2>.
> * we also never put any tag on top of "Define <param2>" like — "the purpose is to compute <param1>"
>
> This is why we said we have **not included** the backward thinking as part of CoT. If you think about it, when “backward thinking” is added the entire generation process becomes “shallower” and requires less deep mental thinking. Without it, the model must use internal states to figure out this “backward thinking” **mentally without writing it down**, such as to mentally derive “A depends on B, B depends on C, C depends on D, and I haven’t computed D yet. Let’s next compute D!”  This is certainly a much “deeper” mental reasoning process and we have certified that the model can learn such operations (when there’s abundant data + transformer is sufficiently deep).
>
> ## Lack of analysis on data quantity
>
> > Lack of analysis on data quantity …
>
> **Answer:** While we can add experiments on this, we claim that it is separate from the main purpose of this paper. It is certainly true that the more data the better accuracy (as well as reasoning length) becomes. It's like Galileo’s Pisa Tower experiment, where we care more about in a controlled setting how two experiments compare; and the height of the Pisa tower may not matter much.
>
> One thing to point out though, is that such high quality of data may not have appeared in GPT-4’s training corpus yet, and that’s why GPT-4o can fail even on iGSM data with op>=11.  To some extent, we are looking into the future — suppose we have abundant data, then what’s the limit of the transformer architecture? Is it capable of performing some “deep reasoning” such as determining the next parameter to compute (via a topological sort)? We believe this can help us understand LLMs better and also better prepare for the future.

---

> ### Author Response · Authors · 2024-11-30
> **Response to Reviewer UB35 [2/2]**
>
> ## Limitation of specific CoT form:
>
> > This paper trains the GPT-2 model using synthetic data with a specific form of CoT. The fune-tuning and probing experimental conclusions are based on such CoT. However, the CoT that is now used to train the LLMs on grade school math problems deviates considerably from this form. This limits the generalizability of the paper's conclusions.
>
> **Answer:** To properly address your concern, we think we have to carefully delve into what "generalizability" means in this context.
>
> To us, it means for this (and similar) logic-following type of reasoning, as long as training data is sufficient and of high quality, and model is sufficiently deep, then transformer is **capable of learning** to perform such logic following, and the inner working **follows our probing**.
>
> If by "generalizability" you mean whether the finding is generalizable to LLMs trained on internet data, then please note, today’s LLMs have not yet been trained on such high quality of data and that’s why they fail on iGSM even with op>=11. Follow-ups (such as 2408.16293) tried simpler formats such as “Box A contains a Box B in it and itself weights 7lb” and GPT-4o continues to fail. To this extent, GPT-4o has not yet learned to accurately perform such logic-following reasoning and thus it's **not interesting to probe GPT-4o** even if we can. However, we verified that the types of errors that GPT-4o fail (see Fig 17) exactly coincide with how our pretrained GPT-2 fail, so this is some kind of generalizability.
>
> If by "generalizability" you mean whether our data can be used to train better all-purpose LLMs, then probably _not directly_. This paper tries to predict the future (after internet data runs out) and studies the limit of the transformer and what’s the data/CoT format needed to achieve such performance, etc. While for today’s models, it appears easy to make improvements --- say can use GPT-4 generated problems to train Phi-3 --- but this will reach a bottleneck; otherwise GPT-4 can improve to GPT-5 already. To overcome this, one had better understand more inner workings about LLMs to do things **less blindly and more scientifically**. This can teach us for instance what’s the data/CoT format needed, and what's the type of reasoning data needed, etc.
>
> With this in mind, what you pointed out, namely **CoTs used by today's models deviate considerably from this form**, is actually an excellent discovery; because we are searching for what type of program-generated synthetic data can improve LLM's reasoning. While our iGSM looks a lot simpler than IMO, powerful models like GPT-4o still largely fail on it -- so iGSM is actually one such starting example of program-generated data (with CoT) applicable even for the state of the art models.
>
> If by "generalizability" you mean whether our methodology is generalizable to other tasks, then it depends on what "others" mean.
> * If it's any logic-following / logic induction type of reasoning then it does generalize --- just a formatting issue and LLMs are superb at learning formats.
> * If it's logic-following reasoning PLUS other aspects, such as (A) common-sense knowledge and (B) arithmetics, then all other issues about (A) and (B) need to be taken into account together. For instance, (A) has issues such as the reversal curse or knowledge manipulations (see 2309.14402), and (B) has issues that one had better reverse the digit orders for better arithmetics. There’s rich literature to study each individual problem (in a synthetic manner) and we expect that we need to gather all the findings from individual studies to eventually help build the next generation of LLMs.
> * If it's some more complex forms of reasoning — such as those needed to prove a hard math problem, we are not there yet. We believe that requires a few more steps (likely also with controlled experiments) to scientifically study how LLMs can best do “planning” (such as a DFS search, in which the context length issue may also kick in). We generally wish to start from the simplest to understand how LLMs function, and hope to start from here. What do you think of this plan?

---

> > ### Comment · Reviewer_UB35 · 2024-12-03
> > **Feedback**
> >
> > I would like to thank the authors for the detailed response. They did address some of my concerns. I appreciate the authors’ explanation of “generalizability.” However, my main concern is about the **specific form of CoT** used in this paper.
> >
> > After reading your response, I still consider there are two flaws:
> > * The specific form of CoT lacks readability compared to the common CoT.
> > * The conclusions drawn based on this specific CoT may not necessarily apply to other forms of CoT. Especially,  CoTs used by today's models deviate considerably from this form.
> >
> > However, I believe that the paper has done a good job of exploring the internal processes of LLMs in solving the GSM problem. I consider this paper should be accepted by ICLR.  6 is a positive score. I will maintain this score and increase my confidence.

---

> > > ### Author Response · Authors · 2024-12-03
> > >
> > > We sincerely thank the reviewer for boosting the confidence, and also hope that this paper can get it (although currently it sits on the boundary...).
> > >
> > > To respond to your final concerns, yes the form of CoT can impact performance. To give you a few examples:
> > >
> > > * ```Define <param> as X; so X = Y * 3 = 7 * 3 = 21``` --- this is what we used in the paper
> > > * ```Define <param> as X; so X = 7 * 3 = 21``` --- this removes the use of Y and makes the "mental process" harder, because the model must not only know X depends on Y, but also be able to retrieve Y = 7 at the same time.
> > > * ```Define <param> as X; so X = 21``` --- this is even harder, as the model has to do 3 times together in a single token position
> > > * ```Y * 3 = 7 * 3 = 21, so <param> is X = 21``` --- this is even harder: because the model has to be able to retrieve Y without explicitly saying <param> in text.
> > >
> > > From top to bottom, the model needs more and more "mental process" without writing things down, so this can impact performance. For instance, if with the topmost CoT the model can solve iGSM with op=32, then with the bottom CoT the model might only solve op=25.
> > >
> > > In fact, we have **prepared a code base that includes many different CoT variants, including some above**, but haven't released it since we are waiting for the final approvals from all of our authors' affiliations.
> > >
> > > We think it is helpful to study different CoT variants, but that's a bit out of the scope. **Not only** we wish to more focus on the logic-following part (that is to decide what's the next parameter to compute), **but also** it becomes easier to quantify the amount of "mental process" using the _value of op_. In contrast, different variants of the CoT format will impact the amount of "mental process", but it's hard to for instance connect that to the number of transformer layers, etc.
> > >
> > > That's the main reason that we decide **not to include it (different CoT variants)** to make the results cleaner and more quantifiable.
> > >
> > > Thanks again for your support. Truly wish this paper can get it, so we can better focus on the follow-ups, including possibly a journal submission later to include additional CoT format comparisons (which can't seem to fit into a conference paper).

---

### Official Review · Reviewer_yuz7 · 2024-11-08

**Soundness:** 3
**Presentation:** 2
**Contribution:** 2
**Rating:** 6
**Confidence:** 3

**Summary:**

This paper attempts to measure the reasoning ability of models by training on a synthetic dataset based on GSM8k. Among other things, they show that training from scratch on this synthetic dataset of grade school math has some signs of generalization to harder problems.



Edited: score to 6 after reviewer rebuttal.

**Strengths:**

Measuring reasoning via training an LLM from scratch is an interesting direction. There are lots of interesting results and experiments in the paper.

**Weaknesses:**

I think the writing can be improved somewhat. Additionally, I think there are some questions about whether the findings are broadly applicable given a single synthetic eval on GSM8k. In some sense, LLMs are useful because they generalize widely beyond a small set of very specific problems.

**Questions:**

Q: My big question is whether this is a proper use of generalization, especially with LLMs, which are useful primarily because they generalize to very different kinds of data. "Generalization" from a synthetic dataset to a similar synthetic dataset is not nearly as ideal as generalization to a very different kind of dataset.

---

> ### Author Response · Authors · 2024-11-30
> **Response to Reviewer yuz7**
>
> We first would like to thank the reviewer for raising the question.
>
> To answer your question, let us first point out that this paper is not looking at the AI models today, but attempting to help / investigate / predict what AI models will be tomorrow.
>
> In fact, it is still somewhat easy to make improvements on LLM reasoning today — say you let GPT-4o keep generating math problems and train from that (e.g., the success of Phi-3 models), but this may eventually reach a bottleneck (otherwise GPT-4 can use such data to improve itself to GPT-5 already).
>
> To overcome such a bottleneck, we believe, one had better understand more intrinsics about LLMs (such as inner working) to do things **less blindly and more scientifically**. In particular, controlled experiments with synthetic data can help us understand what data/CoT formats are required, and what are the typical errors, etc. In contrast, if one continues to use real-world data blindly and hope for the best, then it is likely to make false claims (see for instance the claimed success of the Reflection models, https://huggingface.co/mattshumer/Reflection-Llama-3.1-70B).
>
> Specifically to our submitted paper, we designed data by removing arithemtics and common sense (factual knowledge), in order to study logic flows in a controlled manner using the iGSM dataset. As we said in Line 533-539 in our Conclusion, we are not suggesting that one should directly use iGSM to improve GPT-4 to get GPT-5. Instead,
>
> * We are exploring for instance, in Line 447-453 of Section 4, that for simple logics, it is not necessary to prepare CoT formatted text such as "because I want to compute X, but X depends on Y and Y depends on Z, so let me compute Z first”, and we explained that the reason is due to the depth of LLMs in Section 6. This can help us better understand what type/format of data is more necessary for LLM pretraining and get us better prepared for tomorrow’s LLMs.
>
> * Additionally, using such controlled data one can more directly investigate WHY models make mistakes, and what internal states have caused such mistakes, and help us make predictions about what future LLM training may look like. This is hidden in our Section 5, and we have deferred to appendix (as well as a separate paper) due to space limitation.
>
> * And many more other benefits.
>
> Our results are definitely generalizable to reasoning of similar logics (namely, in context logic following) and it is only a formatting difference. Instead of iGSM, you can design tasks such as “Box A contains a Box B in it and itself weighs 7 lbs” and witness that GPT-4o make similar mistakes as GPT-2 does on the iGSM dataset, see 2408.16293. But
>
> * How about generalizing to similar reasoning logics PLUS other aspects, such as (A) common-sense knowledge and (B) arithmetics? Then all other aspects about (A) and (B) need to be taken into account together. For instance, (A) has issues such as the reversal curse, and (B) has issues that one had better reverse the digit orders for better arithmetics. There’s rich literature to study each individual problem (in a synthetic manner) and we expect that we need to gather all the findings from individual studies to eventually help build the next generation of LLMs.
>
> * Are we capturing more complex forms of reasoning? Of course not yet. But please understand, in contrast to most works that study LLM reasoning in a black-box fashion, we are already making a step forward. We have tried to study one of the most basic forms of reasoning, and believe this is an important first step.
>
> Finally, if you're asking whether our **probing results** apply to real-life models, then our answer is No if you’re concerning models of today but will be Yes for models of tomorrow.  Today’s models such as GPT-4o and Llama3 can still fail quite badly on iGSM data (complete failure on op>=11) or even its simpler data variants (see follow ups such as 2408.16293). Therefore you won’t expect good probing results unless perhaps op<=7, but that’s too simple a problem not worth probing. (Likewise, the logic in problems in GSM8k are too simple and not worth probing.)
>
>
> Thanks and we sincerely hope this has somewhat addressed your main question.

---

> > ### Comment · Reviewer_yuz7 · 2024-12-01
> > **Thank you**
> >
> > Thank you for your response. I've decided to bump my score to 6.

---

### Meta-Review · Area_Chair_bRiL · 2024-12-21

**Metareview:**

Summary of the paper: This paper investigates the mathematical reasoning capabilities of LMs, particularly focusing on grade-school math problems. The paper develops a synthetic dataset called iGSM, designed to enhance template diversity and reduce contamination, enabling controlled experiments. Through elaborate data generation and probing methods, they explore several critical questions lie in LMs: whether models can generalize reasoning skills, replicate human-like problem-solving methods, and how model architecture influences reasoning abilities. Results indicate that LMs, specifically GPT-2, can indeed generalize reasoning processes and learn complex dependencies. The paper finds that these models can solve math problems similarly to humans, engaging in backward thinking processes rather than mere memorization. The study also examines the nature of reasoning mistakes, providing insights into the models' limitations and areas for potential improvement.

Strengths of the paper:
- Innovative Methodology: This paper presents a novel approach to understanding the reasoning capabilities of LMs. Its originality and scientific rigor address an important problem in LLM research. This method advances our understanding of how these systems process and reason about information.
- Systematic Experimentation: By implementing an extensive framework of data synthesis, fine-tuning, and probing techniques focused on grade-school math problems, the study demonstrates a rigorous experimental design. Notably, the decision to train a GPT-2 model from scratch using synthetic data enhances experimental validity and effectively avoids contamination issues that often affect other benchmarks in the field.
- Valuable Insights: The investigation into the LMs' hidden reasoning processes yields important insights into the often opaque workings of neural LMs. The paper presents valuable findings regarding the relationship between model depth and width, the model's generalizability to unknown tasks, and its ability to learn backward reasoning processes. These contributions are important for the future design of language models.

Weaknesses of the paper: I do not have significant concerns about this paper. While many reviewers have raised issues regarding the generalizability of the proposed method—specifically concerning the training and test datasets (iGSM), the size of the LM (GPT2), and the format of the CoT, etc., I believe the authors provided satisfactory clarifications during the rebuttal process. After reading the paper, I consider it a solid starting point for understanding how LLMs function. It is important to begin with simpler approaches to build a foundational understanding.

Reasons for the decision: After carefully considering all the comments, I believe that most concerns have been adequately addressed by the authors. The issues raised primarily revolve around conceptual questions rather than the need for additional empirical experiments. It appears that many reviewers are at least partially convinced by the authors' explanations. All reviewers agree that this paper should be accepted at ICLR. Personally, I find the paper very appealing, particularly the concept of "control experiments" and the overall study framework. This work offers a solid foundation for understanding the reasoning process of LMs in a systematic way. It has the potential to engage the broader community and make a significant impact.

**Additional Comments On Reviewer Discussion:**

As mentioned above, the issues raised by the reviewers primarily revolve around conceptual questions rather than the need for additional empirical experiments. The authors provide clarifications to each issues and reviewers are at least partially convinced by the authors' explanations.

---

### Decision · Program_Chairs · 2025-01-22

Accept (Poster)